# General practitioner and nurse practitioner attitudes towards electronic reminders in primary care: a qualitative analysis

Elizabeth Cecil  ,[1,2] Lindsay Helen Dewa  ,[3,4] Richard Ma,[1] Azeem Majeed,[1] Paul Aylin[1]

[1]Department of Primary Care and Public Health, Imperial College London, London, UK
[2]School of Life Course Sciences, King's College London, London, UK
[3]Patient Safety Translational Research Centre, Imperial College London, London, UK
[4]School of Public Health, Imperial College London, London, UK

**Correspondence to**
Dr Elizabeth Cecil;
e.cecil@imperial.ac.uk

## ABSTRACT

**Objectives** Reminders in primary care administrative systems aim to help clinicians provide evidence-based care, prescribe safely and save money. However, increased use of reminders can lead to alert fatigue. Our study aimed to assess general practitioners' (GPs) and nurse practitioners' (NPs) views on electronic reminders in primary care.

**Design** A qualitative analysis using semistructured interviews.

**Setting and participants** Fifteen GPs and NP based in general practices located in North-West London and Yorkshire, England.

**Methods** We collected data on participants' views on: (1) perceptions of the value of information provided; (2) reminder-related behaviours and (3) how to improve reminders. We carried out a thematic analysis.

**Results** Participants were familiar with reminders in their clinical systems and felt many were important to support their clinical work. However, participants reported, on average, 70% of reminders were ignored. Four major themes emerged: (1) reaction to a reminder, which was mixed and varied by situation. (2) Factors influencing the decision to act on reminders, often related to experience, consultation styles and interests of participants. Time constraints, alert design, inappropriate presentation and litigation were also factors. (3) Negative consequences of using reminders were increased workload or costs and compromising GP and NPs behaviour. (4) Factors relating to improving users' engagement with reminders were prevention of unnecessary reminders through data linkage across healthcare administrative systems or the development of more intelligent algorithms. Participants felt training was vital to effectively manage reminders.

**Conclusions** GPs and NPs believe reminders are useful in supporting the provision of good quality patient care. Improving GPs and NPs' engagement with reminders centres on further developing their relevance to their clinical practice, which is personalised, considers cognitive workflow and suppresses inappropriate presentation.

## INTRODUCTION

Good quality healthcare is safe, effective and patient centred.[1] Primary care administrative systems offer the opportunity to improve

### Strengths and limitations of this study

► Our use of in-depth qualitative interviews has produced novel and varied insights into general practitioners' (GPs)' and nurse practitioners' attitudes toward reminders in electronic health records.

► The views and experiences we collected from GP and nurse practitioners in varying positions based in practices of two regions (North West London, and Yorkshire and Humber) with varying demographic populations has provided a working hypothesis that, we feel, is transferable across GP and nurse practitioners working within the English National Health Service .

► We limited researcher bias by using a peer-reviewed topic guide, having two researchers to validate coding and codevelop the thematic map.

► However, we do acknowledge that parts of the knowledge 'map' may be missing. Primary care clinicians who engaged with the study may have different views than those who did not.

patient care by delivering reminders to clinicians at the point of care. An electronic reminder is a prompt created within a clinical administrative system, through the interrogation of recorded patient data. They remind the clinician of an action required, which may include a recommendation on safe medication prescribing (warning of possible interactions); a more cost-effective drug; advice on a preventive or clinical intervention (such as smoking cessation or immunisation); or to document clinical information (such as blood pressure) (table 1).

There is evidence that clinician behaviours and processes of care provision can be improved by point of care computer reminders, but improvements tend to be small and the features of reminder systems that are associated with clinically worthwhile improvements are difficult to determine.[2] However, in primary care, reminders are

**Table 1** Types of reminders in primary care clinical computer systems

| Reminder type | Software | Purpose | Viewed | Action required | Modal* |
|---|---|---|---|---|---|
| Drug price comparison alert | Prescribing decision support software, for example, ScriptSwitch and OptimiseRx. | To save on costs of prescribed drugs. | A pop-up viewed when prescribing, suggesting a cheaper alternative | Action is required to close the pop-up. Information can be ignored. | ✓ |
| Drug interaction/monitoring alert | GP system standard template. Based on UK pharmaceutical reference publications. Algorithm draws on patient prescription data. | To promote safe prescribing. | A pop-up viewed when prescribing warning of potential drug interactions | Action is required to close the pop-up. Information can be ignored. | ✓ |
| Allergy alert | GP system standard template. Algorithm draws on patient clinical data. | To promote safe prescribing. | A pop-up viewed when prescribing warning of potential allergies | Action is required to close the pop-up. Information can be ignored. | ✓ |
| Diagnosis alert | Diagnosis decision support reminder. for example, sepsis. | To increase awareness of condition as a possible diagnosis. | A pop-up viewed when entering diagnosis-related patient symptoms. | Data must be entered to close the pop-up. | ✓ |
| Quality care standard reminders (eg, quality outcomes framework, influenza vaccine) | GP system templates and toolkits; algorithms draw on patient clinical data. | To promote evidence-based care and maximise financial payment for performance. | On the patient home page and viewed when patient record is opened. Information continues to appear as icons when records are updated. | Most reminders do not have to be actioned and can be ignored. | × |
| Additional patient information alerts from set reminder | Personally created from set template, for example, safeguarding children template (Royal College of General Practice). | To communicate patient-specific information to other clinicians. To create a registry. | On the patient home page and viewed when patient record is opened to those granted access. | No action required when viewing reminder. | × |
| Additional patient information from generic reminder | Personally created from a generic template or free text, for example, highlighting a patient with additional needs. | To communicate patient-specific information to oneself or to other clinicians and practice staff. | On the patient home page and viewed when patient record is opened to those granted access. | No action required when viewing reminder. | × |

*Modal alerts are those where the system shuts down and becomes unobtainable behind the alert. The user must interacted with the alert in order to return to the patient record.
GP, general practitioner.

believed to be particularly important. This is because within the consultation, time pressures, limited information and distracting cues mean that clinicians' thinking (general practitioners (GPs) and nurse practitioners (NPs)), needs to be fast and intuitive rather than systematic.[3] Consequently, there has been an increasing use of reminders for a range of clinical tasks (table 1).

There are barriers to clinical engagement with the reminders. One such barrier is 'alert fatigue',[4] a potential unintended consequence of the increased number of reminders. Alert fatigue happens when the system user becomes desensitised to the digital reminders, and subsequently fails to engage with them. A literature review (predominantly from the USA and within hospital settings) that investigated reminders safety in medication prescribing found between 49% and 96% of alerts were overridden.[4] In 2002, a UK cross-sectional study of primary care clinicians assessed attitudes towards drug interaction reminders and found one in four (of 236 responses) admitted to frequently overriding drug interaction alerts without properly checking them.[5] Other barriers include lack of user friendliness in the administrative systems; time constraints to address the reminder;

and clinicians being unable to see the value in the information provided were all barriers to the acceptance of reminders.[6–10]

Most research in this area has been outside the UK,[8 11 12] where health systems, such as primary care gate-keeping role, can differ.[13] While studies have examined the development of computerised systems to improve quality in primary care,[14 15] studies tend to be quantitative. An up-to-date understanding on how GPs and NPs interact with these systems within the UK is lacking. This study aims to address this gap using semistructured interviews to explore GP and NPs attitudes towards electronic reminders in primary care to support their enhancement.

## METHODS
### Design
We conducted a qualitative study using semistructured interviews. The use of semistructured interviews was chosen because the methodology allows a flexibility to the interview process, which results in the collection of data relating to GP and NPs' attitudes to reminders in electronic health records.

## Participants

We recruited a purposeful sample of GPs and NPs from National Health Service (NHS) general practices located in two regions in England: North West London (NWL), and Yorkshire and Humber (Y&H) between 12 March and 19 September 2018. These regions were selected to obtain opinions from participants serving populations with varying demographics (eg, rural compared with urban patients). We selected participants with experience of primary care administrative systems, based on their position, experience and practice population demographics. Only GPs and NPs were approached as they deliver the majority of care within NHS primary care; other members of the primary healthcare team such as administrators and receptionists are not normally expected to deal with clinical alerts.

In UK, GPs can be self-employed (GP partner) or salaried. We approached partners, salaried and trainee GPs. NPs, trained specialist nurses, have prescribing rights and are likely to face similar challenges, when providing good quality care, as GPs, yet they may have very different attitudes towards the reminders they experience.

Participants were approached through regional National Institute for Health Research (NIHR) Clinical Research Networks (CRN).[16] A researcher (EC) followed up with participants who showed an interest and invited them to take part in the study. Recruitment, interviews and data coding took place simultaneously. The number of subjects required for the study was dependent on the data. Recruitment ceased when researchers felt no new major themes were immerging. Sixteen clinicians were approached, 1 GP (NWL) did not respond and 15 (94%) took part.

## Semistructured interviews

A single researcher (EC) conducted the face-to-face semistructured interviews in a setting of the participant's choice. In most cases (13/15) this was at the participant's practice. The topic guide and participant information sheet were developed through literature[17 18]; meetings between (EC) and patients with lived experience; external peer review; and was piloted with a single GP based in North West London (online supplemental file 1). The topic guide was structured to explore three main areas: (1) perceptions of the value of the information provided in the reminder; (2) reminder-related behaviours; and (3) how to improve reminders. Participant responses were followed up with probing questions for additional depth and detail. Interviews lasted between 45 and 60 min and was audio-recorded with participant's consent. Participants were paid for their time, which was a set amount determined by CRN guidance.

## Data analysis

Interviews were anonymised, transcribed verbatim and subjected to a thematic analysis.[19] We went through six stages: (1) familiarisation with data; (2) generating initial codes; (3) searching for themes; (4) reviewing themes;

(5) defining and naming themes; and (6) interpretation of themes through consultation with the first, second and third authors (EC, LHD and RM). Two researchers (EC and LHD) independently coded (stage 2) three transcripts. Eighty per cent of codes defined by LD mapped to those defined by EC. Codes that did not map were further examined and recoded if necessary. Codes were then grouped into subthemes, and a thematic map (stage 5) was then developed by researchers (EC, LD and RM). We used QSR NVivo V.12 to manage data.

## Patient and public involvement

We presented the study proposal to a small group of five people, identified as having various health conditions and experience in primary care, as part of an existing advisory group. The patient representatives were reimbursed for their time according to NIHR INVOLVE guidance. The representatives provided feedback on the appropriateness of the research design, research questions and the topic guide. Their comments were incorporated in study development. For example, the patient representatives felt it was important to ascertain competency in using reminders (how confident they felt in using electronic health records and whether they thought they had received adequate training to manage the reminders) as this is likely to be a barrier to reminder engagement. These questions were added to the topic guide. A single patient representative supported interpretation of the study findings.

## RESULTS

### Participants

Fifteen participants were interviewed; eight were female (53%), and two-thirds (67%) were GP partners (table 2). Eight participants used SystmOne (53%) and seven used Egton Medical Information System (EMIS) clinical administrative system (47%). Participants' years of experience within primary care ranged from less than one to 34 years with a median of 18 years. Participants' practices were on average larger than national practices, median population (IQR) 9355 (4655 to 14 219) versus 6681 (4065 to 10 034). They were, on average, slightly more affluent: median Indices of Multiple Deprivation scores (IQR) 19 (13 to 28) versus 22 (14 to 32) (where higher scores indicate higher deprivation). Twenty per cent of practices (2/10) located within rural populations compared with 15% nationally (1248/8248).

### Study themes

Participants were familiar with reminders in patients' electronic health records. They reported that reminders were very common and that virtually every patient had at least one occurring during a consultation. The number of reminders varied between patients and were dependent on the patient's health status and time of year.

Our study identified 847 codes within our data that were categorised into subthemes and four main themes:

**Table 2** Characteristics of interview participants and the administrative systems they use

| Attribute | Detail | Number | % |
|---|---|---|---|
| Practicing region | Y&H | 6 | 40 |
| | NWL | 9 | 60 |
| Position | GP partner | 10 | 67 |
| | Nurse practitioner | 3 | 20 |
| | Salaried GP | 1 | 7 |
| | GP trainee | 1 | 7 |
| Gender | Male | 7 | 47 |
| | Female | 8 | 53 |
| Experience (years) | 0–1 | 1 | 7 |
| | 2–10 | 2 | 13 |
| | 11–20 | 5 | 33 |
| | 30+ | 7 | 47 |
| Administrative system | EMIS | 7 | 47 |
| | SystmOne | 8 | 53 |

EMIS, Egton Medical Information System; GP, general practitioner; NWL, North West London; Y&H, Yorkshire and Humber.

**Table 3** Major themes relating to general practitioner and nurse practitioner views of electronic reminders

(1) Reaction to a reminder
 Initial reaction to a reminder
 Dependent on situation
 Acceptance
 Use of reminders to organise consultation
(2) Factors influencing decision to act
 Situational factors influencing decision to act on reminders
 Inappropriate timing of reminder
 Time constraints
 Oversensitive or invalid reminders
 Individual factors influencing decision to act on reminders
 The perceived value of the advice given
 Beliefs reminders promote better care
 Experience in professional role
 Consultation style
 Consequences of ignoring
(3) Consequences of using reminders
 Negative consequences of using reminders
 Costs associated with software
 Changes clinicians' behaviour
 Workload increases
(4) Factors relating to improving reminders
 Need to improve number and validity of reminders
 Improving reminder design
 Making number of reminders manageable
 Standardisation of reminders
 Need to improve efficiency managing reminders
 Training
 Team working
 Patients' role in managing their conditions

(1) reaction to a reminder (2) factors influencing the decision to act on reminders (individual and situational), (3) unintended consequences of using reminders and (4) factors relating to improving reminders (table 3 and online supplemental file 3). Theme examples within the interview data are shown in online supplemental file 2.

### Reaction to a reminder

Participants' initial reaction to a reminder varied. Although participants found reminders frustrating, there was an overall feeling of acceptance towards reminders. Participant's emotional response did, however, depend on the reminder and the situation.

> If it's a busy clinic, stress, because it's, like, 'Ugh.' If it looks like it's going to be a complicated patient and it's an out-of-hours reminder, then relief probably.
>
> GP partner, Y&H, SystmOne

### Use of reminders to organise the patient consultation

Participants reported reminders had become a routine part of their work; they assessed reminders before a consultation and used the information as a guide to organise and prioritise the patient's needs. They used the reminders to build management plans and to relay information to themselves or colleagues. Participants talked of feelings of satisfaction, accomplishment and being in control after actioning reminders.

### Individual factors influencing the decision to act on reminder
### The perceived value of the advice given

Engagement with the reminders varied. Most participants said they would acknowledge the reminder even if they

did not act on it. The majority commonly stated '70%' were ignored.

> I wouldn't be able to do anything if I didn't ignore the majority of them.
>
> GP partner, male, Y&H, EMIS

Generally, participants valued the information provided by the reminders.

> I don't think anybody's ever complained about the serious alerts, it's the infuriating pop-ups.
>
> GP partner, female, NWL, EMIS

Some of the participants reported that their views on reminders had changed over time, from feeling overwhelmed to acceptance as they became more familiar

with them. Most participants valued the advice given by most reminders.

Some participants felt that prompting for potentially overlooked diagnoses was particularly helpful in the primary care setting. This was due to the high patient turnover and short consultation time; these participants believed recording patient information was automatic, while processing the information was sometimes more difficult.

Some participants mentioned current reminders were user-friendly, and most believed they were evidence based, which meant they could keep up to date with guidance, ultimately improving the care they provided. However, a few participants did not share these views and found all reminders intrusive. Overall, the least popular reminders were the drug cost-saving reminders. Participants complained that the price difference between the previously prescribed drug and its alternative was often minimal.

### Experience and professional role

Participants' feelings towards reminders were often dictated by their professional role, clinical interests or years of clinical practice. For example, one reported that he provided out-of-hours care to patients outside his normal practice and that the most important reminders were those that conveyed additional patient information because they supported continuity of care. The trainee GP and NPs indicated that drug interaction reminders were particularly useful in supporting safe delivery of care.

> Everyone's the same, their interests or their increased fields of knowledge are the ones that they're sort of going to look at more.
>
> Nurse practitioner, female, NWL, EMIS

Many participants accepted that colleagues may have different attitudes towards reminders. For example, they suggested that younger clinicians may find managing reminders easier because they tended to be more computer literate. Yet other participants felt older colleagues would find managing reminders easier as they had more experience to process, act on or ignore the information provided, and nurses were generally more likely to action on reminders because they are less likely to take clinical risk decisions.

> … I think doctors are more confident to ignore the prescribing advice than I would be at all.
>
> Nurse practitioner, female, NWL, SystmOne

There was concern by more clinically experienced participants that reminders undermined their competence. Others felt that GP partners would be more likely to act on the reminders that had financial rewards. However, some felt that by focusing on these reminders, it may make patients more sceptical of the care being provided.

> I think the partners will really try and crack through the things more. One could argue it's because money is involved.
>
> GP Partner, female, NWL, EMIS

Participants generally felt the responsibility of actioning a reminder lay with the clinician who was with a patient. However, some seemed to feel more emotionally responsible towards managing the reminders. One participant talked about the guilt associated with failing to action a reminder.

### Consultation style

Consultation styles differed across the participants. Some used the computer to support their consultation, while others felt that it was important to focus on the patient and only enter data after the patient had left the room. Those who did not use a computer reported that they often missed reminders until it was too late.

> Writing up oh I should have done that doesn't matter then whether I'm on time or 20 minutes late because they've still gone out the door.
>
> GP partner, male, Y&H, SystmOne

### Consequences of ignoring a reminder

Participants generally felt they had genuine reasons for ignoring reminders and would document these decisions.

> I would defend – if I got picked up on it, and I wouldn't often – I would be able to defend why I ignored it for that patient.
>
> Nurse practitioner, female, NWL, SystmOne

The consequence of ignoring a reminder differed by reminder type. For example, participants reported that ignoring drug cost-saving reminders resulted in (usually small) financial loses while ignoring drug or allergy reminders could result in a patient safety issue.

> A lot of them are about money, so ignoring … instruction to use a cheaper drug… you might get a rebuke…. Obviously, if it's about patient safety - and a lot of them are about patient safety - but if it feels a true risk and you ignored it then the consequence is that you would do the patient harm.
>
> GP partner, male, Y&H, EMIS

Most participants reported there were also opportunistic consequences. For example, a consequence was failing to notify young women of the need for a smear test that could be considered a missed opportunity as these patients rarely visit their GP.

Some participants' decision to act was influenced by concerns of the consequences of ignoring a reminder. For instance, their behaviour could be highlighted through audit, or ignoring a reminder could lead to a litigation case. NPs felt that they may not get the same support in the case of litigation as GPs and they believed this made them more risk-averse when managing reminders.

I'm very risk-averse, and I'm always aware of litigation …. I'm not convinced that the MDU [Medical Defence Union] would defend me in the same way as they would a GP….

Nurse practitioner, female, Y&H, SystmOne

No participants had experienced significant events after ignoring a reminder, but a few participants knew others who had.

### Situational factors influencing the decision to act on reminders
#### Number of reminders
Participants felt that there were too many reminders, which led to irritation or stress. Factors identified to increase the number of reminders were patient age, comorbidities or time of year. Information was also repeated. For example, if a patient had more than one comorbid condition then a reminder to take blood pressure may be listed for each condition

No, but you end up just clicking cancel 100 times, it just irritates you in the consultation, because you're typing, kind of examining patients, typing, you've got all this like pop up, pop up, pop up, it's just frustrating

Salaried GP, female, NWL, SystmOne

#### Time constraints
Participants reported they had limited time to manage all reminders in a typical consultation partly due to numbers but also because reminders were time-consuming to action.

#### Invalid, irrelevant or oversensitive reminders
Participants believed some reminders were irrelevant, did not improve the care they provided or were purely a tick box exercise. Participants also felt irritated that there were errors in some of the information provided.

But I know that a lot of those warnings are wrong. They're just simply wrong.

GP partner, male, Y&H, EMIS

Participants considered some reminders were too sensitive, for example, recording hay fever in the patient's notes could trigger a sepsis alert. In some cases over sensitive reminders were removed.

Yeah, it's gone, I think it was flawed. It was coming up on virtually everything so I don't know who decided to switch that off

Nurse practitioner, female, Y&H, SystmOne

#### Inappropriate timing of reminder
##### The reason for the patient visit
Participants questioned the appropriateness of the reminders and the need for opportunistic management during the consultation.

It's embarrassing, you've got a patient who's coming in who looks like they're in pain or stress and the first thing you have to do is ask about smoking.

GP partner, female, NWL, EMIS

#### Within the consultation
Participants believed the timing of the alert could be made more appropriate to the consultation's natural progression.

It would be nice if the reminder came up at the appropriate time, so if you are doing a history … there was a history based reminder, if the medication one came up in the medication, when you press examination maybe all the examination ones so that height, weight, blood pressure came up at that time so that, you know, you could do it in a more natural flowing.

GP partner, female, NWL, EMIS

Participants felt that when reminders appeared when the clinician was not with a patient (eg, drug interaction or drug alternative reminders when managing repeat prescriptions), this was annoying.

#### In comparison with the last reminder
Participants talked of how the time of year influenced when reminders appeared, for example, Quality Outcomes Framework indicators are reset in April. Many felt algorithms could be developed so the reminder's timing was more meaningful and took into consideration when the previous reminder was actioned.

#### Changing systems
Participants spoke of frustration in the way systems constantly changed and that not enough precaution was taken to ensure new reminders were fully functional or user friendly.

### Negative consequences of using reminders
#### Reminders are changing behaviour
Participants indicated that reminders have altered GP behaviour within the consultation. It was felt that GP training teaches patient-centred care yet some reminders drive behaviour based on financial incentives. Participants felt they and others could become too reliant on reminders. There was also concern that reminders can force an inappropriate action, recording of incorrect information or make them wary of the information they recorded.

So, because the CCG [Clinical Commissioning Group] will be looking at our referral letters we should get patient's consent to allow them to look at it, that virtually never happens but you have to tick the box

GP partner, male, Y&H, SystmOne

… they do change the way you do things. So I gave you the example of the sepsis reminders. It makes you wary of putting certain information in because you know it's going to prompt a reminder.

GP partner, male, Y&H, SystmOne

 Cecil E, *et al. BMJ Open* 2021;**11**:e045050. doi:10.1136/bmjopen-2020-045050

## Costs and benefits

The participants felt that the reminders suggesting alternatives to cut costs were apparent to the patient and damaged the relationship between patient and clinician. Several participants commented that the cost of the price comparison software probably outweighed any savings.

## Increasing workload

Participants pointed out that the alternative drugs prescribed were often unavailable; this required the patient to come back, which increased their workload.

## A need to improve the number and perceived value of reminders

### Making the number of reminders more manageable

Most participants felt reducing the number of reminders was important. Participants outlined that numbers could be limited by keeping reminders relevant; by removing duplications, which occur when patients have more than one comorbid condition; and by removing drug interactions or price comparison alerts for repeat prescriptions and joining up IT systems across the whole of health and social care to prevent duplication of work.

> Where we're missing information is where things are done outside the practice so, you know, if we had information that was more freely available within the consultation that would prevent duplication, errors, all sorts of things.
>
> GP partner, female, NWL, EMIS

Some participants commented that reminders would be less frequent and more relevant if the algorithms used to trigger the reminders were more 'intelligent'.

### Standardising and providing more context to a reminder

Participants felt that more context could be given to some reminders. For example, if a patient had previously been advised to have a smear test or a blood pressure measurement is requested because of a high reading at the last visit, it would save time going through the notes but also duplication of work. Participants also felt that more use of reminder templates could standardise data recording and help the clinician work more systematically in a consultation.

### Improving the design of reminders

Participants repeatedly called for an improvement in the ease of coding to avoid entering text information. Participants suggested the use of hierarchical drop-down lists, for example, with a sepsis alert, lists could speed entering data on the underlying condition.

> Most participants did comment that the alert designs made some reminders harder to ignore than others (eg, red colour coded reminders).
>
> They flash and they're big, red angry ones.
>
> GP partner, male, Y&H, EMIS

However, the participants had become accustomed to the format of the reminders so some questioned the importance of a redesign. Colour coding or ranking reminders was felt to be too simplistic to highlight important reminders but helpful to define the type of reminder or where the reminder fitted into the progression of a consultation. Participants did feel that redesign of reminders needed to be more individualised to a clinician's consultation style.

> I think, ideally, they should be individualised so people consult in different ways, so if I use the computer in a way that I'm not really entering much until the patient has left the room, they don't work for me in that way.
>
> GP partner, male, Y&H, SystmOne

## A need to improve the efficiency in managing reminders

### Working as a team

The majority of participants felt that reminders were well managed across the practice team, although some did feel that improvements could be made. For example, they suggested receptionists could be encouraged to offer longer appointments to patients with a high number of quality standard reminders or have dedicated staff to manage them.

> It's so difficult to be an acute care clinician plus chronic disease manager at the same time, and that's where alerts sometimes probably get missed or not missed, but consciously avoided. So, an appropriate kind of management of the resource in terms of thinking, right, do we need to have more acute care clinicians or chronic disease managers?
>
> GP partner, male, NW London, EMIS

### Training

All participants commented that new starters, within their practice, received training on using the clinical systems. However, participants felt that the provision of training was variable, and many participants felt they needed more training to increase efficiency and to standardise data entry, particularly when new reminders were introduced.

> To be honest with you, I think we could all do with training. Like when they update, things like that, I think it would be nice to have a session to say there are recent updates, can all the doctors come and catch up?
>
> GP partner, female, NW London, EMIS

> I don't think it was long enough, the training, because actually EMIS is your… well, computer system is how you do everything
>
> GP trainee, female, NW London, EMIS

### Patient self-management of conditions and data

Participants believed that patients needed to take more responsibility for their own health. They indicated that managing reminders often required signposting patients and believed that if patients were more aware of their own needs, they could organise their own care. Some participants also believed that patients' access to amend

records would have a positive impact on the number of reminders as updating information (eg, smoking status and QOF reminder) or organising their annual reviews would reduce numbers. Participants also felt that patients would become more aware of drug cost-savings and agree to switch to a cheaper brand if the reminders were available to patients.

## DISCUSSION
### Key findings
Participants were familiar with the reminders in their clinical systems and felt many were important, providing up-to-date, evidence-based information. Participants' emotional response to the appearance of a reminder depended on the reminder and the situation. Factors influencing the decision to act on the information provided in a reminder related to situational factors (time constraints, appropriateness and reminder sensitivity) and to individual clinician factors (perceived value of advice, GP experience and consultation style). There were negative consequences of using reminders. For example, concerns were reported in becoming over-reliant on the reminders or being forced to record incorrect information. GPs and NPs' views on improving reminders were related to, first, a need to improve the efficiency of managing reminders and second a need to reduce their number and improve their value.

### Comparison with previous research
Our study found similar barriers compared with previous studies.[7–10] These were time pressures in primary care; irrelevant or inappropriate alerts within the clinical workflow; a lack of standardisation of reminders; and a need for training to effectively manage the reminders. Clinicians interviewed in England particularly about quality standard reminders[20] discussed the loss of continuity of care and the increase in workload, but enablers were also reported including more job satisfaction and improved disease management. A study investigating attitudes and motivation of GPs and NPs to deliver public health programmes found attitudes towards the reminders were mixed. Some participants in the study thought reminders were bothersome, irritating and many switched them off, but others felt they facilitated a consultation, and the structured templates ensured essential clinical information was documented.[21] A systematic meta-review (2015)[22] of 11 systematic reviews suggested four significant challenges that needed to be addressed before clinical decision support systems (which include diagnosis alerts) could fully support the diagnosis. The systems needed to: (1) be adaptable and incorporate new knowledge; (2) apply a standardised approach; (3) be deeply integrated with the electronic health record to trigger appropriate timing of the reminder; and (4) and have an understanding of the cognitive workflow of the user. Interestingly, these findings from using a meta-review methodology closely relate to findings from our qualitative research.

### Limitations
In-depth qualitative interviews can offer novel and varied insights into the thoughts and behaviour of participants. We have collected views and experiences from GP and NPs in varying positions based in practices of two regions (NWL and Y&H) with varying demographic populations. Due to the richness of the data collected, the focused research question, the openness of the participants to discuss their views and the saturation of coding into themes, we feel 15 participants was an adequate sample size. We, therefore, feel our findings provide a working hypothesis that is transferable across all GPs and NPs within the English NHS. However, we do acknowledge that parts of the knowledge 'map' may be missing. Primary care clinicians who engaged with the study may have different views than those who did not.

We did not interview practice managers or receptionists who may also be involved in managing reminders within the practice; however, our study focused on obtaining clinician's attitudes to reminders that are specifically delivered at the point of care. We also did not include practice-based pharmacists who are increasingly being employed in GP surgeries to help manage long-term conditions or provide advice to patients on multiple medicines. All participants were from practices using either EMIS Web or SystmOne administrative systems. Although these are the market leaders, 10% of practices in England do not use these systems.[23] However, all primary care administrative systems use the same clinical coding system, prescribing classification, appointment system, consultation classification and core alerts. The systems may slightly differ in the way the reminder is presented and how the clinician interacts with the reminder. For example, some systems may allow the user to 'switch off' the reminder.

We acknowledge that bias could be introduced during the process of data collection, analysis or interpretation. We took steps to limit participant bias by ensuring anonymity and interviewing in a place that was acceptable to the participant. We limited researcher bias by using a peer-reviewed topic guide; having two researchers (LHD and RM) to validate coding and codevelop the thematic map; and involving all authors and a patient representative in interpreting the study findings. Two of the coauthors are practicing GPs (RM and AM), and their interpretations will be informed by practice.

### Implications for future development and policy
Studies have discussed the effectiveness of 'modal' alerts in prescribing[24]; these are reminders where the system becomes unobtainable behind the alert until actions have been completed, forcing the user to input information. However, our participants found these reminders could be the most irritating, particularly if the information they provided was irrelevant or incorrect. Concerns that incorrect information was often being recorded when an action was forced or that users were altering their behaviour to avoid triggering these reminders is worrying and needs addressing. However, these types of reminders

were valued by more cautious or less experienced clinicians. Modal alerts do have their place, in supporting safe practice, if well designed, using 'intelligent' algorithms and users have appropriate training.

A randomised controlled trial, using computer-simulated patients, found early diagnostic suggestions can improve the accuracy of GPs' diagnoses.[25] Our participants commented that processing the information provided by a patient during a consultation could become difficult during busy surgeries and considered reminders to be a useful tool in preventing missed diagnoses.

Our study highlights the need to improve the experience of GP and NPs when interacting with reminders in their patients' electronic health records. Sophisticated use of data to create the reminders; connecting IT systems across the whole of health and social care; and patients' access to their data have all been highlighted as important to progress with the development and improvement of reminders in patient records. The government's target to go paperless[26] and introduce a comprehensive system of electronic health records in England by 2020[27] is a positive move forward in this process. Future development will also need to address different consulting styles used by GP and NPs who will require training to improve the efficiency of managing reminders.

## CONCLUSIONS

GPs and NPs believe reminders are useful in supporting the provision of good quality care. Improving GPs and NPs' engagement with reminders centres on further developing their relevance to a GP's clinical practice, which is personalised, considers cognitive workflow and suppresses inappropriate presentation.

**Acknowledgements** We wish to thank the members of the Patient Safety Translational Research Centre's Research Partners Group for their contribution in the project development and result interpretation. We also thank North West London CRN and Yorkshire and Humber CRN for their support in recruiting participants for this study.

**Contributors** PA secured the funding for this study. All authors contributed to the conception and design. EC secured Health Research Authority approval for the study and conducted all interviews. EC, LHD and RM carried out the analysis. All authors took part in interpreting the findings. All authors commented on and helped to revise drafts of this paper. All authors have approved the final version.

**Funding** This study was funded by National Institute for Health Research (NIHR) Patient Safety Translational Research Centre (PSTRC) programme grant (PSTRC-2016-004). RM is funded by a NIHR Doctoral Research Fellowship (DRF-2017-10-181) for this research project. The NIHR Imperial Patient Safety Translational Centre is a partnership between the Imperial College Healthcare National Health Service (NHS) Trust and Imperial College London. The Dr Foster Unit is an academic unit in the Department of Primary Care and Public Health, within the School of Public Health, Imperial College London. The unit receives research funding from Dr Foster Intelligence, an independent health service research organisation (a wholly owned subsidiary of Telstra). The Department of Primary Care & Public Health at Imperial College London is grateful for support from the NW London NIHR Applied Research Collaboration and the Imperial NIHR Biomedical Research Centre.

**Disclaimer** The views expressed in this publication are those of the authors and not necessarily those of the National Health Service, the National Institute for Health Research, or the Department of Health.

**Competing interests** All authors have completed the Unified Competing Interest form (available on request from the corresponding author), and PA and AB declare that they are partially funded by grants from Dr Foster Intelligence, an independent healthcare information company.

**Patient consent for publication** Not required.

**Ethics approval** Health Research Authority approval was gained (IRAS No: 234951). Ethics approval was not required for this study as participants were NHS staff.

**Provenance and peer review** Not commissioned; externally peer reviewed.

**Data availability statement** Data are available on reasonable request. Deidentified participant data are available, from the corresponding author, on reasonable request. Theme examples within the interview data are shown in supplementary file 2.

**ORCID iDs**
Elizabeth Cecil http://orcid.org/0000-0002-5281-6656
Lindsay Helen Dewa http://orcid.org/0000-0001-8359-8834

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
