## [Reviewer comments · BMJ Open]

ARTICLE DETAILS

TITLE (PROVISIONAL)	General practitioner and nurse practitioner attitudes towards electronic reminders in primary care: A qualitative analysis
AUTHORS	Cecil, E; Dewa, Lindsay; Ma, Richard; Majeed, Azeem; Aylin, Paul

VERSION 1 – REVIEW

REVIEWER	Jeffrey M Weinfeld, MD, MBI Georgetown University Medical Center, USA
REVIEW RETURNED	13-Nov-2020

GENERAL COMMENTS	The researchers sought to understand how UK GPs and PNs interact with alerts and reminders. They used semi-structured interviews and then coded the interviews with thematic analysis. I agree with the patients' concern, that level of EHR competency as well as level of primary care experience (eg trainee, vs PN vs years of practice experience) might effect responses given. I appreciate that this was addressed in the demographics table. However, only one (apparently PN) trainee was enrolled. I'm not sure that this furthers your research question, as trainee attitudes may not be representative of practicing clinicians. Was there consideration to exclusion of trainees? Perhaps more of the trainee comments could be added to the Training section. You describe the modality of typical alerts in Table 1. I suggest using terminology of alerts (modal) vs reminders (non-modal). You mention that distinction in the table and do begin to address under implications. It seems to my reading that most of themes relate to modal alerts but perhaps I'm not correct about that. This would be an interesting observation and helpful to discuss. Finally, from a US perspective, what about the fact that only two EHRs are represented in your sample, even though they are the top UK EHRs? It may be helpful to explain how the use of CDS varies between these systems for an international reader, or to say which system each participant represents. It would be helpful to comment more on which parts of the map you think are missing. Overall, I appreciate the qualitative approach to this topic. This is an important addition to the literature in trying to document the barriers that primary care clinicians face in CDS workflow and in reducing burnout.
---

REVIEWER	Jan Frich University of Oslo
-----------------	---------------------------------

REVIEW RETURNED	27-Dec-2020
-------------

GENERAL COMMENTS	This is a focused and well-written paper. Some minor comments: It is a bit confusing that participant numbers in table 2 differs from N = 15 in this study. Fig. 1 is difficult to read. Could you consider making it simpler or splitting it up or making a table? Page 21: Comparison with previous [rather than "past"] research? References: You need to consult the author hub and format references in accordance with the style (titles of journals need to be abbreviated etc).
--

VERSION 1 – AUTHOR RESPONSE

Reviewer: 1 Prof. Jeffrey Weinfeld, Georgetown University Comments to the Author: The researchers sought to understand how UK GPs and PNs interact with alerts and reminders. They used semi-structured interviews and then coded the interviews with thematic analysis.	
I agree with the patients' concern, that level of EHR competency as well as level of primary care experience (eg trainee, vs PN vs years of practice experience) might effect responses given. I appreciate that this was addressed in the demographics table. However, only one (apparently PN) trainee was enrolled. I'm not sure that this furthers your research question, as trainee attitudes may not be representative of practicing clinicians. Was there consideration to exclusion of trainees? Perhaps more of the trainee comments could be added to the Training section.	We have added a trainee's comments to the training section I don't think it was long enough, the training, because actually EMIS is your... well, computer system is how you do everything [GP trainee, Female, NW London, EMIS] We have added GP trainee comments to our supplementary
You describe the modality of typical alerts in Table 1. I suggest using terminology of alerts (modal) vs reminders (non-modal). You mention that distinction in the table and do begin to address under implications. It seems to my reading that most of themes relate to modal alerts but perhaps I'm not correct about that. This would be an interesting observation and helpful to discuss.	Most alerts are not modal, and themes relate to both modal and non-modal. Modal alerts tend to be drug prescribing alerts. Attitudes towards modal alerts were the most polarised. With less experienced or more cautious clinicians valuing the advice while more experienced clinicians finding the controlling nature irritating. I have updated the text as follows:

		'Concerns that incorrect information was often being recorded when an action was forced or that users were altering their behaviour to avoid triggering these reminders, is worrying and needs addressing. However, these types of reminders were valued by more cautious or less experienced clinicians. Modal alerts do have their place, in supporting safe practice, if well designed, using 'intelligent' algorithms, and users have appropriate training.'
	Finally, from a US perspective, what about the fact that only two EHRs are represented in your sample, even though they are the top UK EHRs? It may be helpful to explain how the use of CDS varies between these systems for an international reader, or to say which system each participant represents.	1) We have highlighted differences in administrative systems in the discussion as follows: 'All participants were from practices using either EMIS Web or SystemOne administrative systems. Although, these are the market leaders, however, 10% of practices in England do not use these systems.²³ However, all primary care administrative systems use same clinical coding system, prescribing classification, appointment system and consultation classification. and core alerts and pop-ups. The systems may slightly differ in the way the reminder is presented and how the clinician interacts with the reminder. For example, some systems may allow the user to 'switch off' the reminder.' 2) We have highlighted the participant EHR system for quotes - EC
	It would be helpful to comment more on which parts of the map you think are missing.	We have added to the strengths and limitations section 'However, we do acknowledge that parts of the knowledge 'map' may be missing. Primary care clinicians who engaged with the study may have different views than those who did not.'P4

		Also updated in discussion
	Overall, I appreciate the qualitative approach to this topic. This is an important addition to the literature in trying to document the barriers that primary care clinicians face in CDS workflow and in reducing burnout.	
	Reviewer: 2 Dr. Jan C Frich, Universitetet i Oslo Avdeling for samfunnsmedisin	
	Comments to the Author: This is a focused and well-written paper. Some minor comments:	
	It is a bit confusing that participant numbers in table 2 differs from N = 15 in this study.	Table 2 has been updated aggregating numbers across Clinician characteristics and administrative systems See page 10
	Fig. 1 is difficult to read. Could you consider making it simpler or splitting it up or making a table?	The figure has been replaced by a table of the themes Map added as a supplementary file
	Page 21: Comparison with previous [rather than "past"] research?	Section heading updated
	References: You need to consult the author hub and format references in accordance with the style (titles of journals need to be abbreviated etc).	References updated